# Can the water resources tax policy effectively stimulate the water saving behavior of social water users? A DSGE model embedded in water resources tax

Zheng Wu[1,2,3], Guiliang Tian[1,2,3]*, Xiaosheng Han[1], Jiawen Li[1], Qing Xia[1]

1 School of Business, Hohai University, Nanjing, China, 2 School of Economics and Finance, Hohai University, Nanjing, China, 3 Jiangsu Research Base of Yangtze Institute for Conservation and High-Quality Development, Nanjing, China

* tianguiliang@hhu.edu.cn

## Abstract

Whether the implementation of the water resources tax policy can stimulate the water-saving behavior of social water users is one of the important criteria for evaluating the implementation effect of the tax reform policy. Taking Hebei Province, the first tax reform pilot in China, as an example. A dynamic stochastic general equilibrium model (DSGE) with embedded water resources tax is constructed to simulate the persistent impact of water resources tax on water-saving objectives. The research shows that: (1) Water resources tax can effectively achieve the goal of water-saving and improve the utilization efficiency of water resources. (2) Levying water resources tax helps to improve the water-saving awareness of enterprises and residents. It can also encourage enterprises to optimize production structures. (3) Rational and efficient use of special water resources protection funds is the basis for ensuring the effective implementation of water resources tax. It can also improve the recycling capacity of water resources. The results show that the government should speed up formulating a reasonable water resources tax rate and accelerate the construction of water resources tax protection measures. To ensure the relatively steady state of water resources utilization and protection, and achieve the dual goals of sustainable economic development and sustainable use of water resources. The research results of this paper reveal the internal logic of the comprehensive impact of water resources tax on the economy and society and provide an important basis for the national promotion of tax reform policy.

**Data Availability Statement:** All relevant data are within the paper. For more detailed data, visit the website of the National Bureau of Statistics of China: http://www.stats.gov.cn/.

## 1 Introduction

With social and economic growth, the uneven spatial and temporal distribution of water resources, frequent water, and drought disasters, human activities interfering with the water cycle, and low water use efficiency are gradually becoming major factors hindering China's green and sustainable development [1, 2]. To address this issue, the government is exploring the development of new water resources management policy initiatives, attempting to shift

**Funding:** This work was supported by the National Social Science Fund Project (Grant No. 19FJYB029).

**Competing interests:** The authors have declared that no competing interests exist.

from levying water resources fees to water resources taxes and establishing a water resources protection tax system in the country [3]. Based on the coexistence of water scarcity and sustainable development requirements in China, water administration departments and scholars generally believe that the current water resources tax design should not consider its fiscal revenue. More attention should be paid to the significance of resource conservation, ecological protection, and green development in water resources [4, 5]. On the one hand, the water resources tax is an effective compensation for using natural resources, which concentrates on the labor value, service value, and ecological value of water resources [6]. On the other hand, the water resources tax effectively curbs the unreasonable demand for water in the region [7], and the tax collection and management is more reasonable and transparent [8]. It improves the efficiency of water use and reduces the exploitation of groundwater, which plays a role in the protection of water resources [9–11], thus promoting environmental protection to a certain extent [12]. In addition, from the various practices of water resources tax in foreign countries, the water-saving effect of water resources tax is also significant [13, 14], but it also has a serious negative impact on agricultural production [15]. Therefore, the implementation of the water resources tax policy needs to be combined with the actual situation of the country to achieve the expected policy effect [16].

In July 2016, Hebei Province, as the first province to actively respond to the national pilot work, put forward the Water Conservation Plan of Hebei Province, in which the water resources tax reform is a significant initiative. Since the implementation of water resources tax reform, it has effectively inhibited the unreasonable water demand in Hebei Province. At the same time, it also promoted the growth of fiscal and taxation in the province. The tax reform has achieved remarkable results, showing that the water resources tax reform work has the basis and conditions for expanding the reform pilot. In November 2017, nine provinces and cities, including Beijing and Tianjin, were added as a new batch of pilot areas to explore further the feasibility of the national promotion of water resources tax reform through pilot projects in different regions. However, most of the 10 pilot areas adhere to the 'Tax and fee for translation' principle to levy water resources tax. Therefore, whether water resources tax can be used as a price lever to alleviate the contradiction between water shortage and economic and social development, and to stimulate the water-saving behavior of social water users, it is still necessary to quantitatively simulate the water resources tax reform policy and evaluate the various economic and social impacts that may be brought after its implementation. This is also an important basic step for the Chinese government to carry out water resources tax reform.

China is in a critical period of water resources tax reform. Chinese scholars have done a lot of research on the applicability and consensuality of the water resources tax system. Wang et al. (2012) proposed that the water resources tax should be administered to match the actual water conditions and geography of the country, and a water resources tax administration model with Chinese characteristics should be established [17]. As the water resources tax pilot work was found, the water resources tax burden standard should be adjusted simultaneously with the change in the local economic level, and the improvement of the ecological environment [18]. The tax revenue collected should be used for special purposes, to enhance the regulating function of water resources tax [19]. However, in the short term, the implementation of a water resources tax policy also harms socio-economic development. Zhao et al. (2021) point out that the water resources tax has increased the tax burden of urban water supply enterprises [20]. Dai et al. (2020) argued that the current water resources tax levy approach had not been applied to agricultural water use, and proposed a farming water resources tax levy and management model that is tailored to local conditions and easy before difficult [21]. Zhao et al. (2021) found that although the introduction of water resources tax has raised the awareness of water conservation, reduced the water demand of enterprises, and improved the efficiency of

water use in various sectors, it has not played a significant role in reducing the total amount of water used in multiple sectors [22]. There are still certain problems in tax collection and administration. However, these studies analyze the implementation of water resources tax reform policy from the macro level, and the implementation effect of a policy also needs to be studied from the micro level. Therefore, as a new tax system, the impact of water resources tax on residents' life, enterprise production, and social development still needs to be studied in depth.

In terms of its characteristics, the water resources tax is a fiscal policy tool. Its regulating effect is not only for the current economic benefit and water-saving effects, but also should be based on the protection of water resources. The policy effect of the water resources tax reform policy should constantly be changing with the development of society, and the DSGE model can meet the current demand for a more comprehensive study of water resources tax. Since Kyland (1982) first used the DSGE model to study the time-series characteristics of the U.S. macro economy [23], the effects of energy price shocks [24], the impact of stochastic technology shocks [25], and coal and carbon resources [26] have been continuously introduced to optimize it. Subsequently, Benavides et al. (2015) used a DSGE model to find that an increase in the tax rate could reduce carbon emissions in the power sector [27]. Still, at the same time, the price of electricity would also increase, and the carbon emission, making carbon emissions reductions and economic growth difficult to achieve simultaneously. Wu (2017) introduced the effects of environmental technologies and energy prices in DSGE to simulate the response changes of the ecological-economic system [28]. Wang et al. (2021) used a DSGE model to conclude that financing conditions can amplify the impact of fuel tax shocks, and the stronger the constraints, the more pronounced the stimulating effect on the economy [29].

Reviewing the existing literature, most of the current studies on the effect of water resources tax policy are short-term or static, and lack of studies on the overall and lasting effects, such as using the CGE model to study the optimal tax rate on water resources [30], which has not yet formed the internal logical structure of the impact of water resources tax policies on economic growth. On the other hand, although the theoretical basis of the DSGE model can solve the above problems, there is a lack of a DSGE model on comprehensive water resources, and the available reference comes from the research results of the DSGE model on carbon taxation. Based on this, this paper constructs a DSGE model incorporating water resources based on the characteristics and economic value of water resources. Taking the first pilot tax reform in Hebei Province as an example to simulate the long-term dynamic response mechanism of water resources tax, to explore how the behavior of micro-entities makes decisions and to evaluate the effect of water resources tax on social water conservation and water resources protection from the perspective of long-term development. The marginal contributions of this paper are as follows: (1) The internal logical structure of the impact of water resources tax reform policy on economic growth is elaborated, and further analyzed the transmission path of the effects of water resources tax on the economy on this basis. (2) A water resource-embedded DSGE model is constructed to analyze the policy effects of water resources tax reform dynamically.

## 2 Theoretical mechanism analysis

In China, water resources are owned by the state. According to the Water Law, units and individuals who directly draw water resources from rivers, lakes, or underground shall apply to the water administrative department or the river basin management agency for a water drawing license and pay water resources fees. The water resources tax reform policy is to change the water resources fee to water resources tax, in the form of a tax, to practice the system of paid

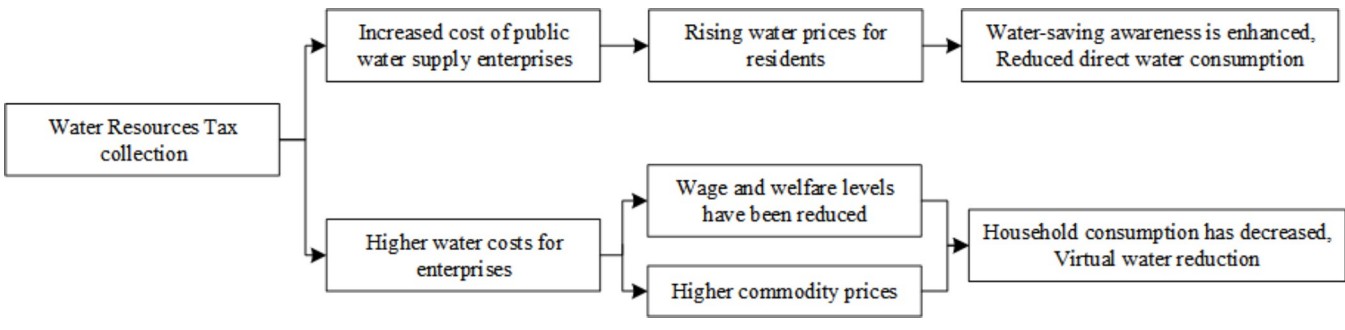

**Fig 1. Influence mechanism of water resources tax levy on residents' behavior.**

use of water resources [31, 32]. The promulgation of the Resource Tax Law of the People's Republic of China also directly shows that the water resources tax is a kind of resource tax [33]. Therefore, the analysis of the effect of water resources tax reform policy conforms to the analysis paradigm of fiscal policy. In this paper, the policy effect of water resources tax refers to the economic impact of water resources tax and the combined effect of the change in the water consumption behavior of water users. Therefore, the analysis of the behavior of micro-actors directly affected by the water resources tax policy implementation can help to understand the transmission mechanism of water resources tax. It can help to understand why each actor chooses to conserve water resources and reduce unreasonable water demand through water resources tax. Moreover, it can help to reflect the way of achieving the structural change of water consumption and the restructuring in water abstraction by the differential tax rate. The analysis of the effect of the water resources tax policy can explain how the water resources tax can alleviate the contradiction between the supply and demand of water resources, change the water supply structure and promote industrial upgrading, as well as infer and argue its impact on economic development. This part analyzes the impact mechanism of water resources tax on residents' life, enterprise production, and social development and draws the impact mechanism diagram (Figs 1–3).

## 2.1 Mechanisms of the impact of water resources tax on people's lives

According to the current pilot tax reform policy document, the water resources tax follows the principle of 'Tax and fee for translation' [34]. Therefore, the reform measures implemented in the current tax reform pilot will have almost no direct impact on residents. However, the rise in water prices is an inevitable trend in the long term. In addition, with the improvement of the water resources tax system and the promotion of the tax reform, residents will gradually realize the importance and urgency of water conservation and water resource protection, thus reducing the unreasonable demand for water and achieving the goal of protecting water resources [35].

As the water resources tax reform is at its early stage, the overall tax rate in each pilot area is still at a low level. Still, the demand for water resources in social development and the scarcity

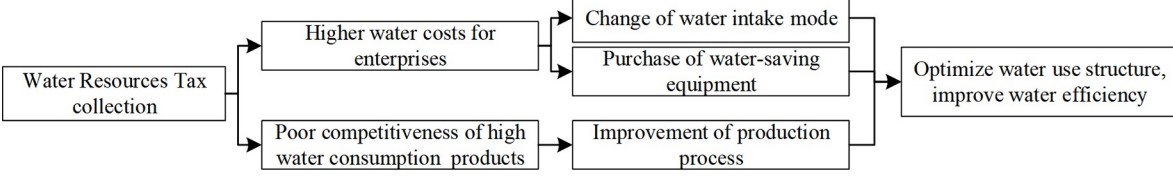

**Fig 2. Mechanism of the impact of water resources tax on enterprise production.**

of water resources put the value of water resources on the rise. Therefore, it is reasonable to consider the increase of the water resources tax rate in this paper, and the water resources tax rate will eventually reach a reasonable range that integrates the supply and demand of water resources and the protection of people's livelihood.

From the perspective of residents as the labor factor, the levy of water resources tax increases the water cost of enterprises. In the short term, it is difficult to achieve either equipment renewal or production plan adjustment, which will reduce the output of enterprises and thus reduce the wages and welfare levels of residents. From the perspective of residents as consumers, in the short term, the increase in production costs of enterprises will be transferred to commodity prices, and the prices of those high-water consumption products or services will increase. At the same time, due to the reduction of income, the reduction of consumption capacity, and the reduction of real disposable money [36], consumers will choose alternatives with lower consumption prices to reduce the consumption of high-water consumption products or services [37]. Consumers are indirectly reducing water consumption by reducing the consumption of high-water consumption products or services. From the virtual water theory perspective, commodities contain the water footprint of the whole production process. With the reduction of consumer commodities, the use of water resources condensed in the commodity itself by the entire production chain is also reduced, which indirectly reduces the consumption of water resources [38]. The mechanism of the water tax's influence on residents' behavior is shown in Fig 1.

## 2.2 The mechanism of the impact of water resources tax on enterprise production

The reverse regulation of the water resources tax reduces the production scale of high water-consuming enterprises, promotes the introduction of water-saving technology, and optimizes

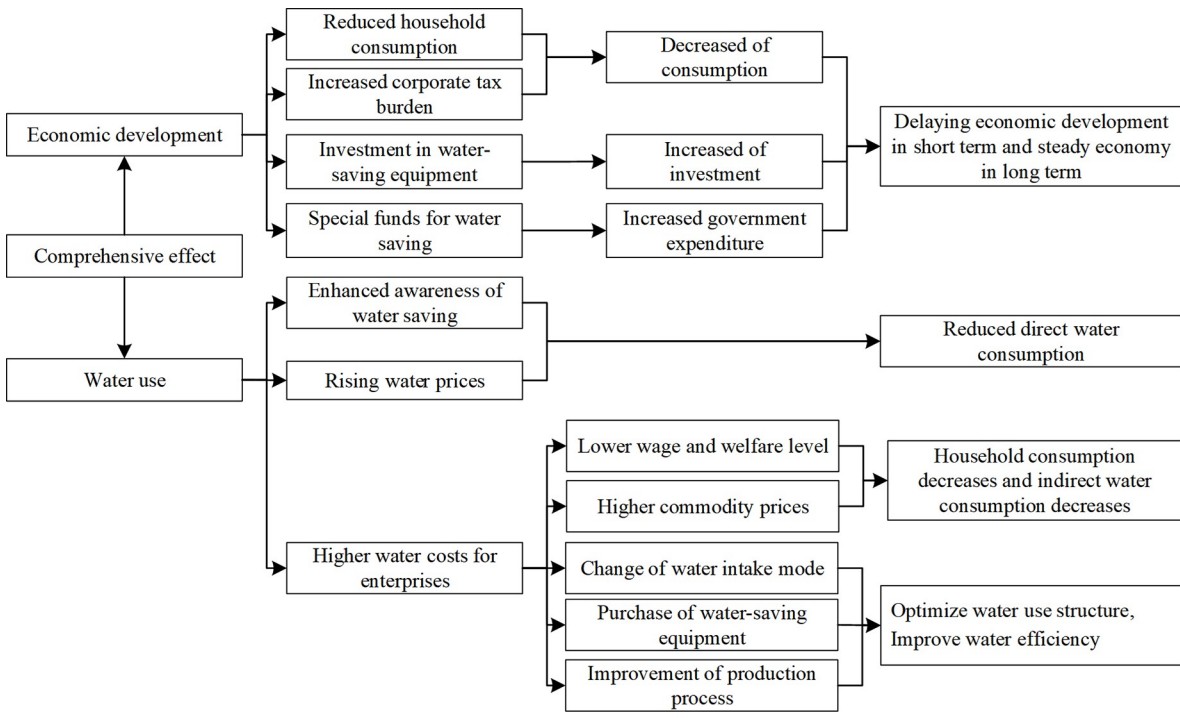

**Fig 3. Integrated impact mechanism of water resources tax on social development.**

the industrial structure [39]. As the current high water-consuming enterprises generally do not use water resources efficiently, the levy of water resources tax will have a more significant impact on high water-consuming enterprises. From the perspective of the production chain, as the production cost of high water-consuming enterprises increases, to keep their profits unharmed, they transfer the tax burden to downstream enterprises or consumers by raising the price of their products in the short term [40], and the resulting impact can be divided into the following two cases.

Case 1: The high water-consuming enterprises produce products that are inelastic in demand, and the transfer of tax burden ensures that their interests are not damaged, because the products produced are in immediate demand, and consumers who buy such products to consume bear the pressure of the tax burden, which reduces the consumption level of consumers and damages the interests of consumers.

Case 2: The high water-consuming firm produces an elastic product, and the price increase makes consumers choose to consume substitutable products for consumption. The higher costs caused by the tax need to be borne by themselves.

Since the current water resources tax is based on the principle of 'Tax and fee for translation' and is an in-price tax, Case 1 is not in line with the actual situation, while the comprehensive result of Case 2 is that high water-consuming enterprises will reduce the scale of production, improve the efficiency of water use, and promote the optimization and upgrading of industrial structure. The positive regulation of water resources tax promotes enterprises to innovate water conservation techniques and guides them to switch from rough to economic production and use of water. Since the water resources tax is a general special tax, the tax revenue is used for the construction of water conservation facilities related to water conservation and the incentive of water conservation effectiveness [41]. On the other hand, in a fully competitive market, innovative water-saving enterprises are rewarded for their low production costs and water-saving production, which increases the market competitiveness of their products and motivates their competitors to transform to water-saving production, and further spreads the impact to the whole production chain. The effect will be transmitted to the entire production chain, optimizing and upgrading the industrial structure and improving water use efficiency [42]. The mechanism of the impact of water resources tax on enterprise production is shown in Fig 2.

## 2.3 Comprehensive impact mechanism of water resources tax on social development

The water resources tax levy will hurt economic development in the short term. At the same time, the economy tends to stabilize in the long term, the efficiency of water use increases, and the value of water use increases. By the national income accounting expenditure method, the driving factors of a country's economy include mainly consumption, investment, and government spending if only closed economy conditions are considered. In the short term, the reduction in consumption occurs instantaneously. It is directly reflected in economic indicators, while the promotion effect of investment on the economy takes a longer time to respond [43]. With the gradual emergence of investment effects, the widespread popularity of water-saving technologies, the improvement of water resources utilization efficiency, and the decline of production water costs [44], the level of wages and benefits, and consumption levels will return to a steady state. At that time, the supply of water resources meets the demand for water resources, and the water ecological environment, the natural cycle of water, and the ecological functions of water bodies are protected. Therefore, the water resources tax reform needs to pay the price of damaged economic development in the short term. From the perspective of long-

term sustainable development, the water resources tax is the most effective means to solve the contradiction between the current water shortage and the growing water demand in China. The mechanism of the impact of water resources tax on economic development is shown in Fig 3.

## 3 Assumptions and model

### 3.1 Assumptions

Dynamic stochastic general equilibrium is a kind of equilibrium state that various economic entities achieve in pursuing their respective goals according to their behavior rules and habits. Therefore, to simplify the complex reality, the DSGE model constructed in this paper is a closed model, including households, firms, and government. The operation of the whole economic system is simulated by constructing the behavior equation of each micro subject. To establish the model equation, it is necessary to put forward corresponding assumptions from four aspects of production, consumption, factor capital, and government departments before constructing the model. Regarding the previous research experience [45, 46] and combined with the research needs of the water resources tax issue, the following four model assumptions are proposed.

1. Assumption 1: On the production side, the market structure is assumed to be a perfectly competitive market, in which manufacturers follow the principles of cost minimization and profit maximization when producing, and the production function has the characteristic of constant returns to scale.

2. Assumption 2: On the consumption side, residents consume various goods according to the principle of utility maximization, and their utility is modeled using the constant relative risk aversion (CRRA) function.

3. Assumption 3: Capital, labor, and water resources are freely mobilizable, while wages conform to the supply and demand theorem and are free to change instantaneously.

4. Assumption 4: The government department only levies water resources tax, and all water resources tax is used for water resources protection management and infrastructure construction expenditure.

### 3.2 Model

The basic framework of this paper is the extension of the research results of Acemoglu et al (2012) [47]. Since there is no research on the application of the DSGE model to the analysis of the implementation effect of water resources tax reform policy, this paper draws on the research method of Yang et al. (2014) on carbon tax [48]. It incorporates water resources as a critical element into the model. Based on the research of Li et al. (2020) [49] and Sun et al. (2021) [50], a water resources embedded DSGE model including households, firms, and the government is constructed.

**3.2.1 Households.** To keep the research subjects consistent, this paper refers to Ireland (2004) [51] to set the behavior of the households sector. Since the water resources tax follows the principle of 'Tax and fee for translation', theoretically, the water resources tax will not directly impact the households sector, so the construction of the households sector follows the classical RBC construction method. Assuming an infinite representative household, the utility of consumption and labor is in the form of CES. To simplify the model, the utility of the

current balance is logarithmic. Thus the utility function of the representative household is:

$$\max_{C_t,N_t,B_{t+1},M_t,K_{t+1}} E_0 \sum_{t=0}^{\infty} \beta^t \left( \frac{C_t^{1-\sigma}}{1-\sigma} - \frac{N_t^{1+\eta}}{1+\eta} + \ln \frac{M_t}{P_t} \right) \tag{1}$$

Where, $E_0$ denotes the conditional expectation factor, $C_t$ denotes consumption, $N_t$ denotes labor, $B$ denotes bond holdings, $K_t$ denotes the amount of capital, $M_t$ denotes the quantity of money, $P_t$ denotes the price level, $M_t/P_t$ denotes real money holdings, $\beta$ denotes the discount factor, $\delta$ denotes the inverse of the intertemporal elasticity of substitution of consumption, and $\eta$ denotes the inverse of the Frisch elasticity of labor supply.

The utility maximization of the residential sector is subject to the constraint that the income in each period is greater than or equal to the expenditure.

$$C_t + (K_{t+1} - (1-\delta)K_t) + \frac{B_{t+1}}{P_t} + \frac{M_t - M_{t-1}}{P_t} \leq \frac{W_t}{P_t}N_t + R_tK_t + (1+i_{t-1})\frac{B_t}{P_t} \tag{2}$$

$$I_t = K_{t+1} - (1-\delta)K_t \tag{3}$$

Where, $W_t$ denotes the nominal wage, $R_t$ denotes the nominal price of capital, and $i$ denotes the bond rate. the investment $I$ that the households sector can decide in the utility function can also be replaced by the amount of capital $K$.

From the utility function and the constraints the Lagrange equation can be constructed.

$$L = E_0 \sum_{t=0}^{\infty} \beta^t \left\{ \frac{C_t^{1-\sigma}}{1-\sigma} - \frac{N_t^{1+\eta}}{1+\eta} + ln\frac{M_t}{P_t} + \lambda_t (w_tN_t + R_tK_t + (1+i_{t-1})\frac{B_t}{P_t} - \right.$$
$$\left. C_t - (K_{t+1} - (1-\delta)K_t) - \frac{B_{t+1}}{P_t} - \frac{M_t - M_{t-1}}{P_t}) \right\} \tag{4}$$

Since money is introduced in the utility function, this paper assumes that the money supply is a non-stationary time series, and The real money growth rate equation is as follows.

$$g_t^m = (1-\rho_m)\log\pi - \log\pi_t + \rho_m g_{t-1}^m + \rho_m\log\pi_{t-1} + \varepsilon_t^m \tag{5}$$

Where, $\pi$ denotes the steady-state nominal money supply growth rate, $\rho_m$ represents the parameter reflecting the growth rate of money supply to inflation. If we use the real money balance $m_t$ and CPI inflation rate $\pi_t$, i.e., $m_t = \frac{M_t}{P_t}$, $\pi_t = \frac{P_t}{P_{t-1}}$.

**3.2.2 Firms.** (1) Introducing the water resources factor into the production function

Since the levy of water resources tax will directly affect the production of enterprises, it is necessary to include water resources as a production factor in the production function. In this paper, regarding the research results of Shao et al. (2013) [52] and Sun (2020) [53], the extended form of the CD function is used to introduce water resources into the production function at the same level of production factors like capital and labor supply, and the equation of the production function is as follows.

$$Y_t = A_t K_t^{\alpha} N_t^{\lambda} Z_t^{1-\alpha-\lambda} \tag{6}$$

Where, $Y_t$ denotes $t$ period output, $\alpha$ is the output elasticity of capital, and $\lambda$ is the output elasticity of labor; $K_t$ denotes capital input in $t$ period; $N_t$ denotes labor supply in period t; $Z_t$ denotes water use in $t$ period; and $A_t$ denotes the technology level in $t$ period, assuming that

technological progress obeys the AR(1) process, yielding:

$$\log A_t = \rho_A \log A_{t-1} + \varepsilon_t^A, \varepsilon_t^A \sim N(0, \sigma_A^2) \tag{7}$$

Where, $\rho_A$ denotes the duration parameter of the technology shock, $\varepsilon_t^A$ denotes the random error under the technology shock, and $\sigma_A$ denotes the standard deviation.

(2) Introduction of water resources tax into the production cost of enterprises

After levying the water resources tax, the taxation department levies according to the number of water resources used, so the amount of water resources tax payable by the enterprise in period $t$ is $T_t Z_t$, where $T_t$ denotes the water resources tax rate in period $t$. To simulate the dynamic effect of the water resources tax, it is assumed to obey the AR (1) process.

$$T_t = \rho_T T_{t-1} + \varepsilon_t^T, \varepsilon_t^T \sim (0, \sigma_T^2) \tag{8}$$

Where, $\rho_T$ denotes the persistence parameter of the water tax shock, $\varepsilon_t^T$ denotes the random error of the water tax shock, and $\sigma_T$ denotes the standard deviation.

The maximization of corporate profits can be expressed as:

$$\max \Pi_t = A_t K_t^\alpha N_t^\lambda Z_t^{1-\alpha-\lambda} - (R_t^K K_t + W_t N_t + T_t Z_t) \tag{9}$$

Where, $R_t^K$ denotes the return on capital.

Because the firm seeks to maximize profit, the optimal first-order condition can be obtained by taking derivatives of $K_t$, $N_t$ and $Z_t$, respectively.

$$R_t = \alpha A_t K_t^{\alpha-1} N_t^\lambda Z_t^{1-\alpha-\lambda} = \alpha \frac{A_t K_t^\alpha N_t^\lambda Z_t^{1-\alpha-\lambda}}{K_t} = \alpha \frac{Y_t}{K_t} \tag{10}$$

$$W_t = \lambda A_t K_t^\alpha N_t^{\lambda-1} Z_t^{1-\alpha-\lambda} = \lambda \frac{A_t K_t^\alpha N_t^\lambda Z_t^{1-\alpha-\lambda}}{N_t} = \lambda \frac{Y_t}{N_t} \tag{11}$$

$$T_t = (1-\alpha-\lambda) A_t K_t^\alpha N_t^\lambda Z_t^{-\alpha-\lambda} = \alpha \frac{A_t K_t^\alpha N_t^\lambda Z_t^{1-\alpha-\lambda}}{Z_t} = (1-\alpha-\lambda) \frac{Y_t}{Z_t} \tag{12}$$

**3.2.3 Government.** In addition to the effectiveness of water savings, the water resources tax policy effect should also focus on what impact its collection may have on socio-economic development. The water resources tax policy effect is the combined effect on economic growth and the water-saving effect of water-saving behavior after optimal choices by a series of actors guided by the government's water-saving objectives. To simplify the model, it is assumed that the source of government tax revenue is the water resources tax and that the income tax revenue is set up as a special fund for water resources protection entirely for water resources protection. The government revenue equation is as follows:

$$G_t = T_t Z_t \tag{13}$$

where $G_t$ denotes the $t$ period government expenditure.

Since water resources are incorporated into the economic system cycle as a factor of production, according to the general equilibrium model idea, water resources need to be resource-constrained to converge to the steady-state, based on this, this paper assumes the water cycle equation.

$$Z_t = (1-\psi) Z_{t-1} + G_t \cdot Z_t / Y_t \tag{14}$$

where $\psi$ denotes the rate of water depletion per period and $Z_t/Y_t$ denotes the amount of water used per unit of output. In the steady state, this equation indicates that the converted value of water resources depletion in the current period will be compensated by the value transfer function of water resources tax in the next period.

**3.2.4 Market equilibrium.** When the market reaches clearing equilibrium, the resource constraint is.

$$Y_t = C_t + I_t + G_t \tag{15}$$

## 3.3 Data sources

The water resources data used in this study are from Hebei Water Resources Bulletin from 2000 to 2020. The data of social and economic indicators such as capital volume and GDP are from the China Statistical Yearbook from 1989 to 2020. Data that cannot be obtained directly from statistical data are estimated according to previous research methods.

## 4 Results

### 4.1 Parameter estimation

**4.1.1 Calibration of structural parameters.** The calibration of capital elasticity parameter $\alpha$ and labor elasticity $\lambda$ can be used to replace capital income indicator by total capital formation and labor compensation by labor income indicator, and the long-term average value of $\alpha$ can be calculated as 0.48 and $\lambda$ as 0.5; the subjective discount parameter $\beta$ can be calculated as 0.975 based on the average value of historical one-year national bond yield of 2.6% of Ying for Finance; the capital depreciation parameter $\beta$ is calculated as 0.975 based on the 1989–2017 China Statistical Yearbook, the ratio of long-term average value of depreciation of fixed assets to long-term average value of capital stock in Hebei Province is calibrated to obtain $\delta$ as 0.07; parameter $\psi$ of water resources depletion rate is obtained as a proxy variable based on the water consumption rate of water resources in Hebei Province Water Resources Bulletin from 2000–2020 as $\psi$ as 0.26; parameter $\sigma$ is set to 1 by referring to the study of Huang (2005) [54]; parameter $\eta$ is set by referring to Hu et al. (2012) [55], calibrated $\eta$ to 3.

**4.1.2 Dynamic parameter estimation.** In this paper, HP filtering is used to obtain the fluctuation components of the observed data, and then the regression is performed by least-squares according to the first-order autoregressive assumption of the shock. The coefficient of the first-order lag term can be used as the persistence parameter, and the standard deviation of the regression can be used as the estimate of the standard deviation of the exogenous shock, taking the natural logarithm of both sides of the output function and making the first-order difference can be obtained as follows.

$$\begin{aligned}
\log A_{t+1} - \log A_t \\
= \log Y_{t+1} - \log Y_t - \alpha(\log K_{t+1} - \log K_t) - \lambda(\log N_{t+1} - \log N_t) - (1 - \alpha - \lambda)(\log Y_{t+1} \\
- \log Y_t)
\end{aligned} \tag{16}$$

The time series data of GDP $Y$, capital $K$, and labor $N$ are substituted into the above equation through Eviews, and the volatility components are obtained through HP filtering. The AR (1) of the tax rate variable $T$ is used as the regression equation to obtain the persistence parameter of the water tax rate.

The regression equation for monetary variable $M$ with AR (1) of first order difference.

$$\log M_t - \log M_{t-1} = (1 - \rho_m)\log\pi + \rho_m(\log M_{t-1} - \log M_{t-2}) + \varepsilon_t^m \tag{17}$$

The persistence parameter of the monetary shock is obtained by the same method.

**Table 1. Regression results for each variable.**

| Variable | First order lag coefficient | Standard error of coefficient | t-Test | p v. | Regression standard deviation |
|---|---|---|---|---|---|
| Tax rate $T$ | 0.517062 | 0.161812 | 3.195455 | 0.0035 | 0.094511 |
| Currency $M$ | 0.545762 | 0.134929 | 4.044821 | 0.0004 | 0.042964 |

The results were obtained using Eviews software operations as follows.

As shown in Table 1, the regression results of dynamic parameters are all significant and plausible. All parameters are shown in Table 2.

## 4.2 Calculation of steady-state values

The steady-state value is generally calculated by first setting $A$ to 1, and then manually solving for some of the steady-state values based on this. In addition, the steady-state value can be set first for some variables, such as labor, assuming a person works 8 hours a day, the steady-state of $n$ can be set first to 1/3. The long-term per capita water use in Hebei province is 275m³/person, and the logarithmic value of 2.44 can be used as the steady-state value of water use $z$. After setting the steady-state value, Dynare is used to calculate the steady-state value of the whole system.

## 4.3 Impulse response analysis of water resources tax shocks

**4.3.1 Responses of enterprises to water resources tax shocks.** Compared with the imposition of water resources fees, the overall price of water in Hebei Province increases after the imposition of a water resources tax. In this context, the scenario of a 1% increase in the water resources tax rate is simulated. As seen in Fig 4, the water resources tax shock causes a short-term decline in water consumption and a gradual return to a steady state in the long term. In periods 1–7, the water resources tax rate gradually returns to a steady state, while the effect of water use suppression gradually diminishes. In periods 8–20, the water resources tax rate is steady, while water use is still suppressed; this suggests that the water resources tax has effectively reduced water use and suppressed water demand for a longer time. In periods 1–7, both the effect of raising water prices and reducing water use is gradually diminishing. The change in water prices is greater than the change in water use and an increase in tax revenues, all of which are used to invest in water conservation and other areas, enhancing the recycling capacity of water resources. In periods 8–20, water price has returned to a stable state, while water consumption is still gradual recovery. Compared with the initial state, tax revenue also shrink, but at the same time, the enhanced recycling capacity of water resources is restored, and the

**Table 2. Summary of parameter setting values.**

| Parameter | Representative meaning | Calibration value |
|---|---|---|
| $\beta$ | Subjective discount factor | 0.975 |
| $\sigma$ | Reciprocal of intertemporal elasticity of substitution of consumption | 1 |
| $\eta$ | Inverted Frisch elasticity of labor supply | 3 |
| $\delta$ | Capital depreciation rate | 0.07 |
| $\alpha$ | Elasticity of capital output | 0.48 |
| $\lambda$ | Elasticity of labor output | 0.50 |
| $\psi$ | Loss rate of water resources | 0.26 |
| $\rho_m$ | Persistent parameters of monetary policy shocks | 0.55 |
| $\rho_T$ | Persistent parameters of water resources tax shocks | 0.52 |

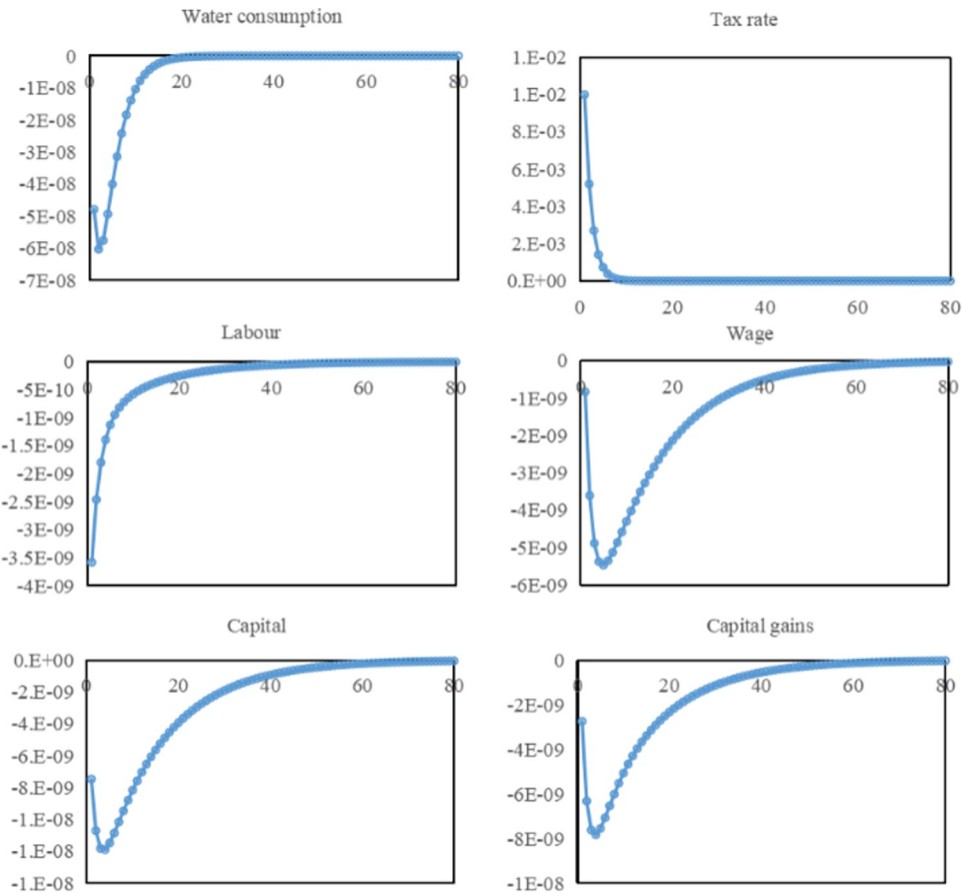

**Fig 4. Responses of enterprises to water resources tax shocks.**

required input also shrinks. Tax revenue is in a relatively stable state. In contrast, water consumption gradually returns to a steady state. Compared with the initial state, water use does not shrink, but the water ecosystem continued restoration of water ecosystems can meet the increasing water demand. In the long run, the increased recycling capacity of water resources meets the growing water demand and ensures sustainable economic and social development.

The impact of the water resources tax reduces labor and lowers the wage level. On the one hand, due to the effects of the water resources tax, the industrial structure is optimized and adjusted, resulting in structural unemployment and reduced labor quantity. On the other hand, production enterprises facing an increase in the tax rate will take measures such as introducing water-saving equipment, upgrading water-saving technology, improving the production process, etc., which brings about the improvement of efficiency, making labor time decrease. In periods 1–5, there is a vertical recovery of labor, some unemployed people find jobs again, and labor time gradually recovers. After period 6, the recovery of labor slows down, and some unemployed people need a longer time to search for matching positions while the labor time of employed workers recovers. With the gradual employment of the unemployed, labor returns to a steady state by period 45. A decline in wage levels accompanies the decrease in labor. On the one hand, the impact of the water resources tax makes it more expensive for firms to produce water, with lower output obtained per unit of labor and lower welfare pay. On the other hand, the reduction in the amount of labor also reduces the wages required to be paid. In periods 1–5, there is a steep decline in wages. Still, due to the gradual recovery of labor

time and the increase in the value of unit labor, the wage level gradually increases after period 6. It returns to a steady state in period 65.

The impact of water resources tax reduces both capital and capital gains. With higher costs, lower output value, and lower profits, the return per unit of capital also decreases, and the nature of capital for profit makes capital investment decreased. However, with the general implementation of the water resources tax policy, the demand for water-saving technology and water-saving equipment has increased, and the prospect of research and development of water-saving technology and manufacturing of water-saving equipment is promising. The investment in such development will gradually increase. In addition, the government supports enterprises to change their production methods and improve their water-saving capacity, encouraging them to purchase water-saving equipment and develop water-saving technology. They will also invest a lot of capital in water-saving. Although the water resources tax affects the capital investment in the short term, in the long term, due to government support and enterprise demand, the capital factor investment will still return to a steady state. In periods 1–3, capital and capital gains continue to decrease simultaneously. And in periods 4–65, capital gains gradually recover due to the positive market outlook for water-saving technologies and equipment, and capital investment also recovers, returning to steady-state levels in period 65.

Compared with the actual social situation in Hebei Province, it is found that from the implementation of the tax reform policy in July 2016 to 2020, the cumulative reduction of groundwater overexploitation in Hebei Province is 4.35 billion $m^3$, the ecological water consumption has increased significantly, and the downward trend of groundwater has been effectively alleviated. Promoting the water resources tax reform policy can effectively promote the regional water-saving effect, and the water ecosystem restoration continues. In addition, the levy of water resources tax led to increased production costs. The major high-water consumption enterprises continue to introduce advanced water-saving equipment and technology, and strengthen water-saving management. The industrial water repetition rate in Hebei Province reached 90.8% in 2020. This also proves that the simulation results of the implementation of the water resources tax reform policy are consistent with reality.

**4.3.2 Resident response to the water resources tax shock.** The shock of water resources tax leads to a decrease in wage levels, a reduction of disposable income of residents, and consequently a decrease in consumption. Higher production costs, lower profits, and lower-wage levels for businesses caused by higher water resources tax. The decline in short-term income of residents as workers directly reduce the level of consumption in the short term, and consumption decreases. From Fig 5, the wage level keeps declining in the periods of 1–5. Industrial restructuring, continuous decline in short-term profits, reduction in capital investment, and increase in the unemployed population all aggravate the decline in wage level. With the tightening of consumer constraints, enterprises have to recover funds in time by lowering product prices, the price level can fall. It increases the natural money balance of residents and the real purchasing power, which promotes the recovery of consumption. In the period of 6–60, the reduction of unemployed people, the improvement of labor efficiency of employed people, the popularization and application of water-saving technologies, the reduction of water costs, and the recovery of corporate profits promoted the gradual return of the wage level to the steady state. With the recovery of wages, disposable income increased, and the consumption level was further restored, returning to a steady state in the period of 60.

The impact of the water resources tax makes residents save more and promotes the increase of long-term investment. Under the influence of water resources tax, the decline of corporate profits leads to the decline of residents' wages, the income of residents in the current period decreases, the opportunity cost of consumption increases, and the propensity of residents to save increases. Under the lower-income level, people tend to worry about future life security,

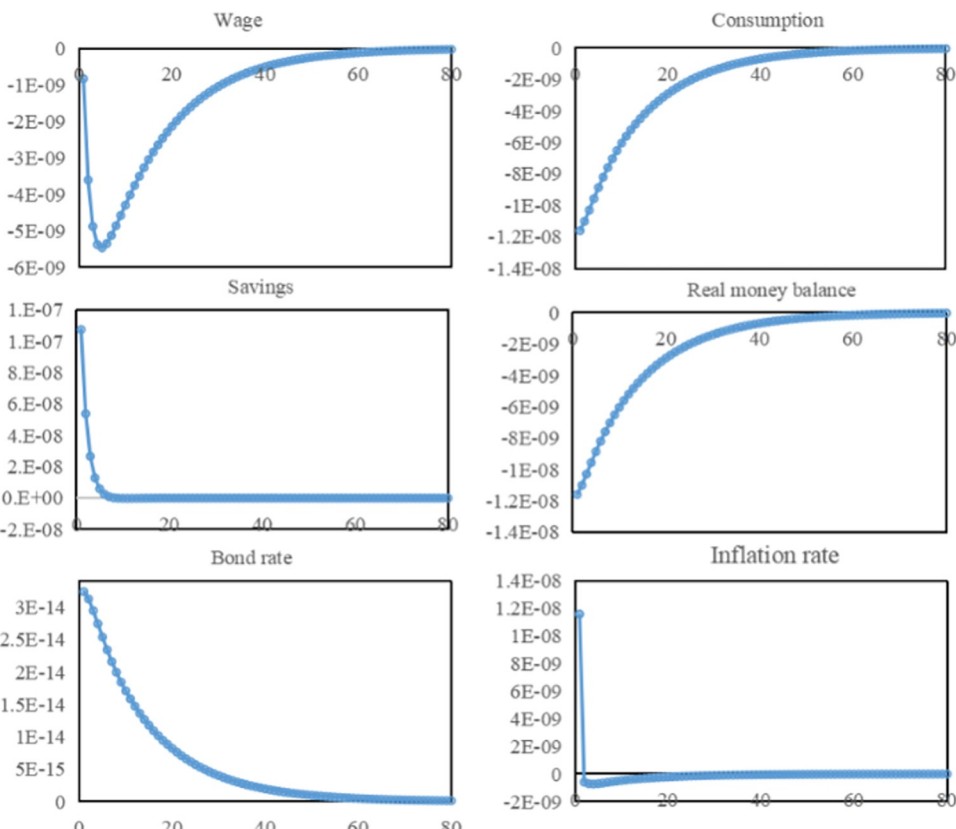

**Fig 5. Resident response to the water resources tax shock.**

curtail unnecessary current consumption and ensure that basic consumption in the future is satisfied. In periods 1–7, the increase in savings gradually slows down. On the one hand, the continuous decline in wage level makes the funds available for saving also shrink gradually. On the other hand, the gradual recovery of consumption makes more funds available for consumption, which weakens the incentive to save. In the 8th period, savings return to a steady state. With the recovery of wages, the increase in income and consumption reach a relative balance, and the share of savings remains unchanged. Since saving funds is equal to investable social funds, saving is the supply of funds, and investment is the demand for funds. Both are two sides of the same coin, so the dynamics of long-term investment should be the same as saving.

The impact of the water resources tax has reduced the natural money balance held by residents. On the one hand, the wage decline reduces residents' disposable income. On the other hand, with the continued impact of the water resources tax, there is a strong demand for water-saving equipment and technologies, which gives companies related to the development of water-saving technologies an incentive to obtain R&D financing by raising bond rates. The increase in bond returns makes the opportunity cost of holding money higher for residents, who will allocate more of their wealth to bond investments. The water resources tax only raises the inflation rate in period 1. In contrast, from period 2 until the steady state, the implementation of the water resources tax, in turn, lowers the inflation rate, raises the real purchasing power, and promotes residents to hold money. But even though the purchasing power of money increases, in the face of future consumption concerns and reduced income, residential consumers still allocate more wealth to savings and bonds with higher short-term investment

returns. From Fig 5, the impact of savings is short-lived, just in periods 1–8, so the most important reason for residents to reduce their money holdings is the higher opportunity cost due to lower incomes. In periods 1–60, the natural money balance remains steady as the bond rate gradually decreases. In the long run, residents' wealth distribution is still determined by their expected profits.

As the standard of water resources tax levied on residents' water use in Hebei Province is 'Tax and fee for translation', the residents of water resources tax will not have a direct impact. However, from the actual situation in China, improving water resources tax collection standards is inevitable. Therefore, the analysis of this part can be used for reference to re-formulate the standards of residential water resources tax in the future.

### 4.3.3 Comprehensive impact of water resources tax on the economy and society

In general, the level of aggregate output measures the overall state of economic development. From Fig 6, the shock of the water resources tax causes the total production of the economy to fall in the short run and return to a steady state in the long run. In the case of a closed economy, total output is composed of consumption, investment, and government spending. Total output briefly rises in period 1, due to the increase in tax expenditures and investment outweighing the decrease in consumption. Still, from period 2 onward, total output enters a phase of sustained decline. Combining the changes in total output fluctuations in the previous periods, it can be assumed that total output has a downward trend in the short run. In periods 2–5, total output continues to decline, the pulling effect of tax expenditures and investment on the economy weakens, and the impact of reduced consumption on total output increases. In periods 6–60, both tax expenditures and investment reach a steady state. Changes in total output are mainly influenced by changes in consumption, with both total output and consumption showing an upward trend. As water-saving technologies become widespread, corporate profits return to normal, wage levels recover, consumption levels rise, total output gradually recovers. In the period 60, total output and consumption return to a steady state.

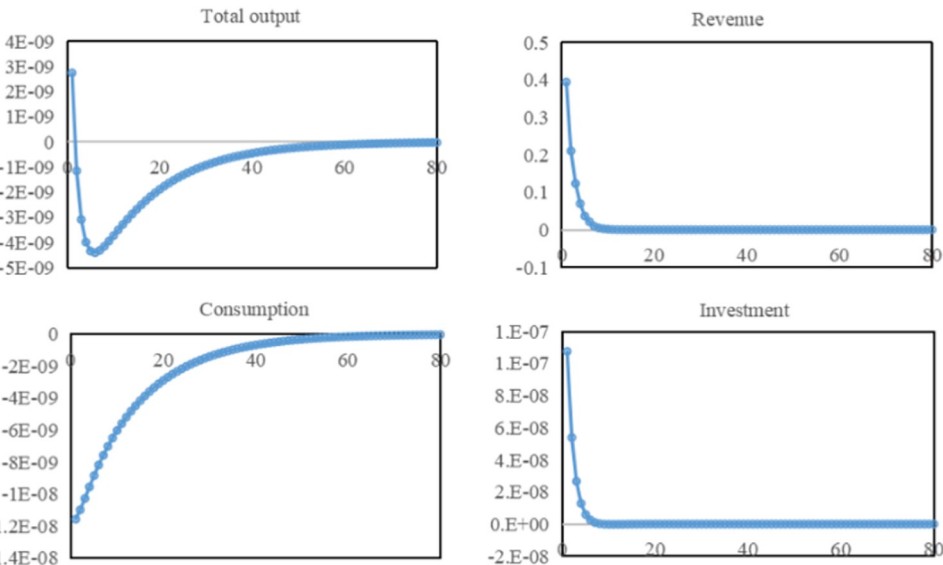

**Fig 6. Overall impact of water resources tax on economy and society.**

Under the water resources tax, water consumption dropped significantly in the short term. Still, the increase in water consumption cost made enterprises' profits drop, resulting in lower wage and benefit levels and less labor quantity and time. At the same time, enterprises introduce water-saving technologies and equipment to improve water use efficiency and reduce production costs to ensure long-term growth and respond to the government's water resources tax policy. In contrast, as workers, residents have seen their savings and money holdings fall, their bond investments rise as they receive less income, and their consumption shrinks. The increase in residents' savings and the rise in bond investment boosted business investment, accelerating the development of water-saving technologies and the diffusion of water-saving equipment. In the long run, due to the enhanced water conservation technology, the cost of water use decreases, enterprises regain normal profits, wages return to normal levels, residents' consumption capacity continues to recover, and socio-economic development returns to a steady state.

Compared with the actual economic situation of Hebei Province, it is found that since the water resources tax was collected, the investment volume, import and export volume, residents and enterprise income of Hebei Province have increased year by year. Still the growth rate decreased compared to before the tax reform. Because in the real social development, many factors affect the investment volume, import and export volume, residents, and enterprise income. Therefore, to a certain extent, it can offset the negative impact of the collection of water resources tax on economic growth and import and export volume, which is also the reason why the level of social and economic development can gradually return to a steady state under the impact of water resources tax. Compared with the water resources fee, levying the water resources tax increases Hebei provincial government revenue. By 2019, the province's cumulative income from water resources tax was 6.941 billion yuan, with an average monthly income of 165 million yuan, which was more than twice the average monthly income from water resources fees in 2015. This also shows that collecting water resources tax can effectively increase government revenue. Still, its tax revenue is not transferred to residents and enterprises, which also affects the economic and social development of Hebei Province to a certain extent. This also proves that the simulation results of the implementation of the water resources tax reform policy are consistent with reality. To make the level of social and economic development return to a steady state as soon as possible, it is necessary to earmark the tax collected and transfer the corresponding payment to residents and enterprises.

## 5 Discussion

The contradiction between the shortage of water resources and the increasing demand for water resources has become one of the fundamental reasons that hinder the sustainable development of the social economy [56]. Optimizing water resource allocation through economic leverage and promoting regional water conservation has gradually become an effective water resources management method [57]. The successful experience of 10 pilot provinces of water resources tax in China also proves that the water resources tax reform policy helps promote the rational utilization and systematic management of water resources in China [58]. Through the quantitative simulation analysis of the water resources tax reform policy, it can also be found that the collection of water resources tax has an obvious incentive effect on the water-saving behavior of social water users. It also significantly impacts the comprehensive development of residents, enterprises, and the social economy.

(1) The collection of water resources tax can effectively achieve the water-saving goal, and help promote the water-saving production of enterprises, change the industrial water consumption structure and improve the utilization efficiency of water resources.

The implementation of the water resources tax reduced the proportion of industrial and agricultural water use. It increased the proportion of ecological and domestic water use while the overall water use remained relatively stable [59]. Since the water resources tax adopts differential tax rates on the way and amount of water drawn by enterprises, the implementation of the water resources tax firstly affects the production of enterprises. It increases the overall production cost of enterprises. While the cost of the living water of residents is not directly affected for the time being but is indirectly affected by changes in income level and commodity consumption. The increase in water cost reduces the number of water resources used by enterprises. Meanwhile, it reduces enterprises' current output and unit labor value, thus reducing the wage level. The decline of residents' income as laborers in the current period directly affects their willingness to distribute wealth: they reduce consumption and hold money and increase savings to guarantee future consumption demand. The decrease in current consumption and abandonment of high-water consumption products further promote enterprises to introduce water-saving technologies and equipment to reduce the cost of water use. The increase in savings provides a stable supply of funds for banks, thus ensuring that enterprises can finance through the difficult transition period. At the same time, tax funds used for water resources protection, on the one hand, can increase the capacity of sustainable water supply. On the other hand, enterprises should be encouraged to develop water-saving technologies, adopt water-saving equipment for production, improve production processes and promote upgrading industrial structures [60]. Through the investment of special tax funds, the production cost of enterprises decreases, the profit returns to the average level, the wage level gradually recovers, the consumption activities increase, and the economic development returns to stability [61]. Overall, the water resources tax reduces industrial water demand and improves water efficiency. Although it hurts economic development in the short term, after long-term recovery, the economy returns to stability, and the sustainable supply capacity of water resources is improved to achieve the dual goals of economic development and water environment protection.

(2) The collection of water resources tax is conducive to improving the water-saving consciousness of enterprises and residents. The price adjustment helps encourage enterprises to change water intake mode and optimize production structure.

Due to the low collection standard of water resources fee and the collection time is not fixed, it often misses the phenomenon, leading to low water-saving awareness among water users [62]. The mandatory water resources tax effectively promotes the necessity of water-saving production in enterprises. In particular, if high-water-consuming enterprises do not pay attention to water saving, it will increase production costs, leading to a decline in enterprise profits, a decrease in market competitiveness, and a reduction in production scale [60]. From the view of long-term development, the transformation and upgrading of enterprise production is the inevitable trend. When the price of water is to a certain level, the cost of water should be paid more attention to. For example, the water price of t Beijing's particular industry is 160 yuan / m$^3$. The high cost of water inhibits the extensive water use behavior of particular industry enterprises, the awareness of water-saving is enhanced, and each water account is actuarially calculated. The implementation and publicity of the water resources tax policy have also improved residents' awareness of water-saving and enhanced the awareness of water scarcity and the urgency of water resources protection [63]. At the same time, the income of enterprises affects the income level of residents, resulting in reduced consumption, especially the consumption of high-water consumption products. For example, due to the increase in car washing costs, the number of car washing is diminished. The choice of consumers forces enterprises to introduce water-saving technology and equipment for water-saving production, improve the production process to promote industrial structure to intensive production

transformation. With the improvement of water-saving awareness of residents and enterprises, water consumption gradually decreases, the utilization efficiency of water resources steadily increases, and the carrying capacity of the water environment consolidates and strengthens. The supply and demand capacity of water resources reaches equilibrium, promoting the economy's sustainable development and water resources environment.

(3) The collection of water resources tax effectively protects the water ecological environment and improves the recycling capacity of water resources and the water supply capacity to meet water demand.

The water resources tax restrains the desire of enterprises to develop and utilize water resources endlessly, ensuring the intergenerational equity of water resource utilization. It provides the necessary means for water resources protection and sustainable utilization. Although the impact of the water resources tax will hurt the economy in the short term, the promotion of the water resources tax is still worthy and must be implemented. Tax funds provide a financial guarantee for local governments to control environmental water pollution, restore water ecological functions [64], and use the funds obtained from production enterprises to repair the ecological damage caused. It also reflects fiscal and tax policies' effect on social wealth redistribution and social equity [65]. Therefore, the rational and efficient use of water resources tax funds is essential for an effective water resources tax policy. Currently, the expenditure items of special funds for water resources tax need to be further improved, the implementation of funds needs to be further supervised, and the corresponding water-saving incentive measures need to be further supplemented [66]. Water resources tax policy requires perfect water on tax means of rigid and a more comprehensive incentive measure covering. The government's correct guidance and water-saving incentives have enhanced enterprises' confidence in water-saving priority. It can be innovative development, accelerate the process of enterprises to complete the introduction of water-saving technology facilities, promote the optimization and upgrading of industrial structure, and improve the utilization efficiency and use-value of water resources. The water resources tax absorbs the profits of enterprises and supplements the ecological environment, which ensures the recovery of water resources recycling capacity and the improvement of water supply capacity under the condition of increasing water demand and realizes the dual goals of sustainable economic development and sustainable use of water resources [67]. The internal relationship of water resources tax policy effect is shown in Fig 7.

## 6 Conclusion

As a price lever, water resources tax is an effective measure to implement the idea of 'water saving priority'. On the one hand, the levy of a water resources tax can change the unreasonable behavior of water use and force enterprises to improve production methods to improve water use efficiency. On the other hand, it can also reduce the dual pressure of the economy and environment so that consumers can establish a correct view of resources and protect water resources spontaneously and consciously. By deconstructing the internal logic of the impact of the water resources tax reform policy on economic growth, this paper puts forward the transmission path of the effects of water resources tax on finances. On this basis, the water resources elements are included in the DSGE model for the first time. The impact of water resources tax on micro-sector and socially sustainable development is simulated through impulse response analysis. It was found that, on the one hand, the water resources tax can effectively achieve the water conservation goal and help promote water-saving production, change the industrial water use structure, and improve water use efficiency. On the other hand, the water resources tax can help raise awareness of water conservation among enterprises and residents. The adjustment of the price mechanism is helpful to encourage enterprises to change the way of

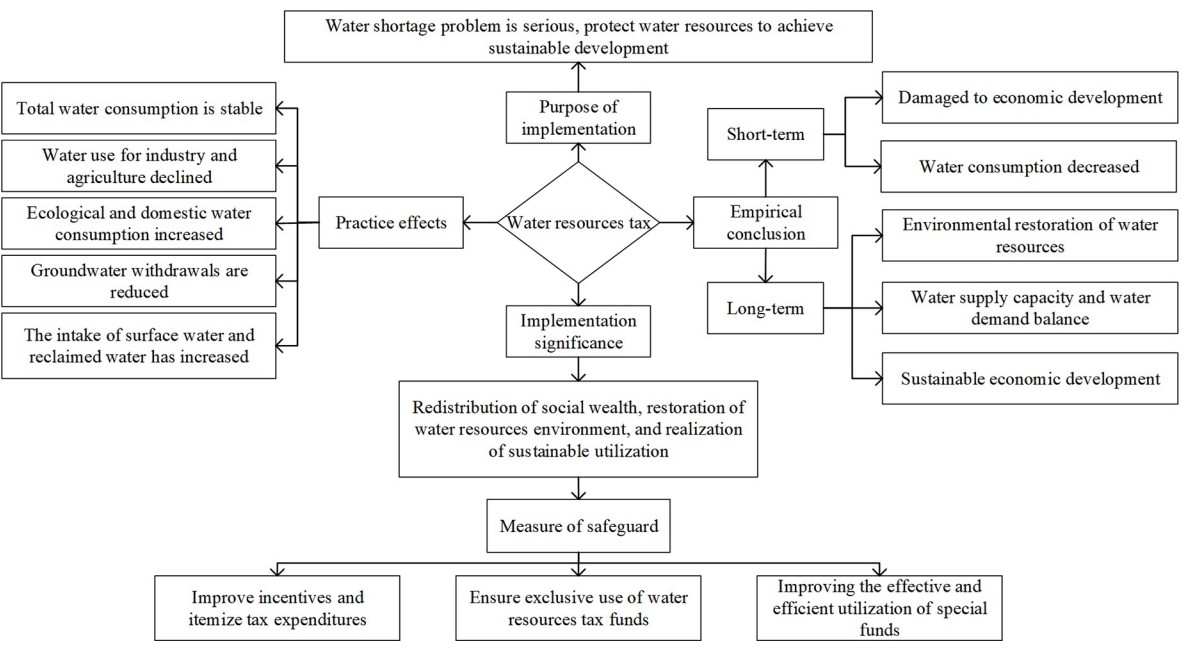

**Fig 7. Internal thread of water resources tax policy effect.**

water intake and optimize the production structure. water resources tax In addition, the effective implementation of the water resources tax is guaranteed by the reasonable and efficient use of special water resources protection funds. In this way, the water ecological environment can be effectively protected, and the recycling capacity of water resources and water supply capacity can be improved to meet the water demand.

The conclusions of this paper provide a theoretical basis for the national implementation of water resources tax reform policy to some extent. When implementing the tax reform policy, the government should speed up the exploration of a reasonable range of water resources tax rates that integrates water resources supply and demand and protects people's livelihood. When implementing the tax reform policy, the government should increase the exploration of a reasonable range of water resources tax rates that integrates water resources supply and demand and protects people's livelihood. Only in this way can people's livelihood be guaranteed while water resources tax is levied and the relatively steady state of water resource utilization and protection can be achieved. Water resources tax revenue should be earmarked for use. On the one hand, rational and efficient use of water resources tax for water resource protection can increase the ability of sustainable water supply. On the other hand, it can encourage enterprises to develop water-saving technology, improve the production process and promote the upgrading of industrial structure. At the same time, the correct guidance from the government to enterprises can enhance their confidence in prioritizing water conservation and innovative development. To some extent, it can accelerate the process of completing the introduction of water conservation technology facilities and achieve the dual goals of sustainable economic development and sustainable use of water resources.

## Author Contributions

**Conceptualization:** Zheng Wu.

**Data curation:** Xiaosheng Han.

**Investigation:** Jiawen Li.

**Methodology:** Zheng Wu, Xiaosheng Han.

**Project administration:** Guiliang Tian.

**Resources:** Jiawen Li.

**Software:** Zheng Wu.

**Supervision:** Guiliang Tian.

**Visualization:** Zheng Wu.

**Writing – original draft:** Zheng Wu.

**Writing – review & editing:** Guiliang Tian, Qing Xia.

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
