## [Decision Letter · Decision Letter 0]

14 Sep 2022

PONE-D-22-20549Can the water resources tax policy effectively stimulate the water saving behavior of social water users?PLOS ONE

Dear Dr. Tian,

Thank you for submitting your manuscript to PLOS ONE. After careful consideration, we feel that it has merit but does not fully meet PLOS ONE’s publication criteria as it currently stands. Therefore, we invite you to submit a revised version of the manuscript that addresses the points raised during the review process.

The authors need to revise the manuscript per the reviewers' comments. 

We look forward to receiving your revised manuscript.

Kind regards,

Abdul Majeed

Academic Editor

PLOS ONE

Journal Requirements:

"We would like to acknowledge the supports of the National Social Science Fund Project (Grant No. 19FJYB029)."

 "This work was supported by the National Social Science Fund Project (Grant No. 19FJYB029)."

"The authors have no conflicts of interest to declare that are relevant to the content of this article."

5. PLOS requires an ORCID iD for the corresponding author in Editorial Manager on papers submitted after December 6th, 2016. Please ensure that you have an ORCID iD and that it is validated in Editorial Manager. To do this, go to ‘Update my Information’ (in the upper left-hand corner of the main menu), and click on the Fetch/Validate link next to the ORCID field. This will take you to the ORCID site and allow you to create a new iD or authenticate a pre-existing iD in Editorial Manager. Please see the following video for instructions on linking an ORCID iD to your Editorial Manager account: https://www.youtube.com/watch?v=_xcclfuvtxQ.

6. Please amend either the title on the online submission form (via Edit Submission) or the title in the manuscript so that they are identical.

7. Please ensure that you refer to Figure 7 in your text as, if accepted, production will need this reference to link the reader to the figure.

8. Please upload a copy of Figure 8, to which you refer in your text on page 11. If the figure is no longer to be included as part of the submission please remove all reference to it within the text.

Reviewers' comments:

Reviewer's Responses to Questions

**Comments to the Author**

1. Is the manuscript technically sound, and do the data support the conclusions?

Reviewer #1: Partly

Reviewer #2: Yes

Reviewer #3: Yes

Reviewer #4: Partly

2. Has the statistical analysis been performed appropriately and rigorously? 

Reviewer #1: No

Reviewer #2: Yes

Reviewer #3: Yes

Reviewer #4: Yes

3. Have the authors made all data underlying the findings in their manuscript fully available?

Reviewer #1: Yes

Reviewer #2: Yes

Reviewer #3: Yes

Reviewer #4: Yes

4. Is the manuscript presented in an intelligible fashion and written in standard English?

Reviewer #1: No

Reviewer #2: Yes

Reviewer #3: Yes

Reviewer #4: Yes

5. Review Comments to the Author

Reviewer #1: The abstract needs to be revised. Some sentences are not grammatically correct (e.g., L18-21). Furthermore, some sentences are too long.

The Introduction of the article does not provide the sufficient information on China’s current water resources tax system and how it is being reformed. While the authors have mentioned several times that the government is trying to reform the existing system, it is not clear how and to what extent. Furthermore, the research gap is not established, and authors merely rely on DSGE model to ‘solve the problems’. However, in fact, the research problem is not fully clarified. Please rewrite the introduction section while keeping in view above points.

I have noticed the similar problems in Section 2. Authors need to clarify what are the existing reforms that are being undertaken/planned. Furthermore, this section suffers from lack of literature support. Please add more literature and citations to support your arguments.

What is the source of Figure 1-3?

Hypotheses 1-4 are not in fact testable Hypotheses but rather are phrased like ‘assumptions’. Please clarify. There is a lot of difference between Hypotheses and Assumptions. In case of assumptions, please explain the logic.

The model which is being developed is not grounded in any theory (or at least it is not explained here). Furthermore, the variable definitions have not been operationalized. No sources have been cited for the equations (unless it is authors’ own work).

The authors have not described data sources on which the analysis is based. It is also not clarified whether authors use simulations to present the results. In case of simulations, different scenarios and corresponding values (initial and final values) are not presented. Furthermore, the simulated scenarios have not been explained to show their real-world implications. This also hasn’t been done in the other sections.

The discussion section lacks any kind of comparison with other studies (domestic and foreign) and fails to summarize and contextualize the findings.

Overall, the author needs to make significant revisions to the article if it is to be published.

Reviewer #2: 1. The author needs to emphasize the innovative value of this paper.

2. The Introduction part should start from the phenomena and problems in practice and lead to the research problem.

3. The literature review should reflect the value of this research.

4. The innovation of this paper and the contribution made by previous studies have not been clearly expressed.

5. This article has obtained some interesting findings through the models, but these findings need to be further verified from theory or actual conditions. Also, further highlight the contribution of this article.

6. Compared with the available literature, what are the theoretical contributions and application values of this study? It is suggested to enhance the corresponding discussions in the conclusion part.

7. The following literature should be helpful for your research：（1）Decoupling economic growth from water consumption in the Yangtze River Economic Belt, China.（2）Coordination of the Industrial-Ecological Economy in the Yangtze River Economic Belt, China.

Reviewer #3: The paper is an interesting study and a timely research work, presenting analysis of the behavior of micro-actors directly affected by the implementation of water resources tax policy that can help to understand and reduce unreasonable water demand through water resources tax. It is a complete work; well written and structured with extensive literature review and comprehensive analyses. The authors should however proofread their paper once more as some parts are still not very clear or need further improvements. Otherwise, excellent contribution to the body of knowledge in the respective field.

Reviewer #4: The manuscript entitled Can the water resources tax policy effectively stimulate the water saving behavior of social water users? is a good contribution to the existing body of knowledge in the related discipline. The paper used dynamic stochastic general equilibrium model (DSGE) embedded in water resources tax to simulate the persistent impact of water resources tax on water saving objectives.

• I found very poor English and sentence structuring throughout the paper. Hence extensive English editing may be required.

1. For example, use of “and” in the first line of abstract more than thrice.

2. Extraordinary long sentences misleading the true meaning of what the authors are trying to explain.

3. Study aim / need of study should be clear in the abstract

4. The objectives of the study are not narrated in proper English

5. It is good if the abstract become a bit short rather such long as the reader may get bored.

• Keywords may be changed

• The theoretical framework and the methodology are the most important and well elaborative part of the study and I want to appreciate the author for such deep insight

• I think there is a lack of references in the discussion section.

• All the results should be balanced with some references with prior studies and may be discussed in the discussion section.

Overall, the paper need improvement in English and is good to be published in the journal.

6. PLOS authors have the option to publish the peer review history of their article (what does this mean?). If published, this will include your full peer review and any attached files.

Reviewer #1: No

Reviewer #2: No

Reviewer #3: No

Reviewer #4: No

---

## [Author Response · Author response to Decision Letter 0]

26 Oct 2022

Dear Editors and Reviewers:

Thank you for your letter and for the reviewers' comments concerning our manuscript entitled “Can the water resources tax policy effectively stimulate the water saving behavior of social water users? A DSGE model embedded in water resources tax” (ID: PONE-D-22-20549). Those comments are all valuable and very helpful for revising and improving our manuscript, with guiding significance to our research. We have studied these comments carefully and have made improvements which we hope will be satisfying. The main improvements in the paper responding to reviewers’ comments are as follows:

Reviewer #1:

1. The abstract needs to be revised. Some sentences are not grammatically correct (e.g., L18-21). Furthermore, some sentences are too long.

Response: As Reviewer #1 suggests, the quality of the ‘Abstract’ should be improved. The logic of the abstract is not strong. Some sentences have grammatical problems, and long sentences affect the reader's understanding of the article. Therefore, we rewrote the abstract and revised the paper's research questions, methods, conclusions, and significance. The detailed revisions are as follows (see also L12-27 of the manuscript).

Abstract: Whether the implementation of the water resources tax policy can stimulate the water-saving behavior of social water users is one of the important criteria for evaluating the implementation effect of the tax reform policy. Taking Hebei Province, the first tax reform pilot in China, as an example. A dynamic stochastic general equilibrium model (DSGE) with embedded water resources tax is constructed to simulate the persistent impact of water resources tax on water-saving objectives. The research shows that: (1) Water resources tax can effectively achieve the goal of water-saving and improve the utilization efficiency of water resources. (2) Levying water resources tax helps to improve the water-saving awareness of enterprises and residents. It can also encourage enterprises to optimize production structures. (3) Rational and efficient use of special water resources protection funds is the basis for ensuring the effective implementation of water resources tax. It can also improve the recycling capacity of water resources. The results show that the government should speed up formulating a reasonable water resources tax rate and accelerate the construction of water resources tax protection measures. To ensure the relatively steady state of water resources utilization and protection, and achieve the dual goals of sustainable economic development and sustainable use of water resources. The research results of this paper reveal the internal logic of the comprehensive impact of water resources tax on the economy and society and provide an important basis for the national promotion of tax reform policy. (Manuscript: L12-27)

2. The Introduction of the article does not provide the sufficient information on China’s current water resources tax system and how it is being reformed. While the authors have mentioned several times that the government is trying to reform the existing system, it is not clear how and to what extent. Furthermore, the research gap is not established, and authors merely rely on DSGE model to ‘solve the problems’. However, in fact, the research problem is not fully clarified. Please rewrite the introduction section while keeping in view above points.

Response: As suggested by the reviewer, the implementation status of China's current water resource tax reform system is an important part of this paper. The research question of this paper should also be condensed from reality. Therefore, we rewrote the introduction part. Added the current situation of the reform, as well as the innovation of this article and contribution. Critical revisions are shown below (see L37-125 of the manuscript for details).

In July 2016, Hebei Province, as the first province to actively respond to the national pilot work, put forward the Water Conservation Plan of Hebei Province, in which the water resources tax reform is a significant initiative. Since the implementation of water resources tax reform, it has effectively inhibited the unreasonable water demand in Hebei Province. At the same time, it also promoted the growth of fiscal and taxation in the province. The tax reform has achieved remarkable results, showing that the water resources tax reform work has the basis and conditions for expanding the reform pilot. In November 2017, nine provinces and cities, including Beijing and Tianjin, were added as a new batch of pilot areas to explore further the feasibility of the national promotion of water resources tax reform through pilot projects in different regions. However, most of the 10 pilot areas adhere to the ‘Tax and fee for translation’ principle to levy water resources tax. Therefore, whether water resources tax can be used as a price lever to alleviate the contradiction between water shortage and economic and social development, and to stimulate the water-saving behavior of social water users, it is still necessary to quantitatively simulate the water resources tax reform policy and evaluate the various economic and social impacts that may be brought after its implementation. This is also an important basic step for the Chinese government to carry out water resources tax reform. (Manuscript: L60-74)

However, these studies analyze the implementation of water resources tax reform policy from the macro level, and the implementation effect of a policy also needs to be studied from the micro level. Therefore, as a new tax system, the impact of water resources tax on residents’ life, enterprise production, and social development still needs to be studied in depth. (Manuscript: L91-94)

The marginal contributions of this paper are as follows: (1) The internal logical structure of the impact of water resources tax reform policy on economic growth is elaborated, and further analyzed the transmission path of the effects of water resources tax on the economy on this basis. (2) A water resource-embedded DSGE model is constructed to analyze the policy effects of water resources tax reform dynamically. (Manuscript: L121-125)

3. I have noticed the similar problems in Section 2. Authors need to clarify what are the existing reforms that are being undertaken/planned. Furthermore, this section suffers from lack of literature support. Please add more literature and citations to support your arguments.

Response: According to the suggestion of this reviewer, we further clarify in Section 2 what the reforms under way are. In addition, we have added relevant literature and applications to support our thesis. Critical revisions are shown below (see L127-227 of the manuscript for details).

In China, water resources are owned by the state. According to the Water Law, units and individuals who directly draw water resources from rivers, lakes, or underground shall apply to the water administrative department or the river basin management agency for a water drawing license and pay water resources fees. The water resources tax reform policy is to change the water resources fee to water resources tax, in the form of a tax, to practice the system of paid use of water resources (Zhang, 2019; Berbel et al., 2019). The promulgation of the Resource Tax Law of the People’s Republic of China also directly shows that the water resources tax is a kind of resource tax (Xiong et al., 2019). Therefore, the analysis of the effect of water resources tax reform policy conforms to the analysis paradigm of fiscal policy. In this paper, the policy effect of water resources tax refers to the economic impact of water resources tax and the combined effect of the change in the water consumption behavior of water users. Therefore, the analysis of the behavior of micro-actors directly affected by the water resources tax policy implementation can help to understand the transmission mechanism of water resources tax. It can help to understand why each actor chooses to conserve water resources and reduce unreasonable water demand through water resources tax. Moreover, it can help to reflect the way of achieving the structural change of water consumption and the restructuring in water abstraction by the differential tax rate. The analysis of the effect of the water resources tax policy can explain how the water resources tax can alleviate the contradiction between the supply and demand of water resources, change the water supply structure and promote industrial upgrading, as well as infer and argue its impact on economic development. This part analyzes the impact mechanism of water resources tax on residents' life, enterprise production, and social development and draws the impact mechanism diagram (Figure 1-3). (Manuscript: L127-146)

4. What is the source of Figure 1-3?

Response: Figures 1-3 are summarized based on the analysis in Section 2. We didn't make it clear in the original manuscript. Therefore, we thank the reviewers for their suggestions. We illustrate the source of Figures 1-3 in L144-146 of the manuscript.

This part analyzes the impact mechanism of water resources tax on residents' life, enterprise production, and social development and draws the impact mechanism diagram (Figure 1-3). (Manuscript: L144-146)

5. Hypotheses 1-4 are not in fact testable Hypotheses but rather are phrased like ‘assumptions’. Please clarify. There is a lot of difference between Hypotheses and Assumptions. In case of assumptions, please explain the logic.

Response: According to the suggestion of this reviewer, we took a closer look at this section. As the reviewer stated, this section should be ‘assumptions’. Because the economy and society are complex and changeable. Therefore, it is necessary to put forward corresponding assumptions from four aspects of production, consumption, factor capital, and government departments when using the DSGE model for simulation. We made changes where appropriate. (L230-238 of the manuscript)

Dynamic stochastic general equilibrium is a kind of equilibrium state that various economic entities achieve in pursuing their respective goals according to their behavior rules and habits. Therefore, to simplify the complex reality, the DSGE model constructed in this paper is a closed model, including households, firms, and government. The operation of the whole economic system is simulated by constructing the behavior equation of each micro subject. To establish the model equation, it is necessary to put forward corresponding assumptions from four aspects of production, consumption, factor capital, and government departments before constructing the model. Regarding the previous research experience (Zhang et al., 2020b; Dissou et al., 2012) and combined with the research needs of the water resources tax issue, the following four model assumptions are proposed. (Manuscript: L130-138)

6. The model which is being developed is not grounded in any theory (or at least it is not explained here). Furthermore, the variable definitions have not been operationalized. No sources have been cited for the equations (unless it is authors’ own work).

Response: Many thanks to the reviewers for their suggestions on the model. We neglected these details in the initial manuscript. We have modified the model section to add the basis for our model development (L252-257 of the manuscript). At the same time, some explanations of variables are added, and the corresponding literature and citations are added (L258-337 of the manuscript).

The basic framework of this paper is the extension of the research results of Acemoglu et al (2012). Since there is no research on the application of the DSGE model to the analysis of the implementation effect of water resources tax reform policy, this paper draws on the research method of Yang et al. (2014) on carbon tax. It incorporates water resources as a critical element into the model. Based on the research of Li et al. (2020) and Sun et al. (2021a), a water resources embedded DSGE model including households, firms, and the government is constructed. (Manuscript: L252-257)

7. The authors have not described data sources on which the analysis is based. It is also not clarified whether authors use simulations to present the results. In case of simulations, different scenarios and corresponding values (initial and final values) are not presented. Furthermore, the simulated scenarios have not been explained to show their real-world implications. This also hasn’t been done in the other sections.

Response: According to the suggestion of this reviewer, we add the data sources of the article in Section 3 (L338-342 of the manuscript). As the reviewer said, the simulation results should be compared and analyzed with the actual situation further to illustrate the practical significance of the simulation results. Therefore, we have added the corresponding content in Section 4 (L433-442, L488-492, L523-539 of the manuscript).

3.3 Data sources 

The water resources data used in this study are from Hebei Water Resources Bulletin from 2000 to 2020. The data of social and economic indicators such as capital volume and GDP are from the China Statistical Yearbook from 1989 to 2020. Data that cannot be obtained directly from statistical data are estimated according to previous research methods. (Manuscript: L338-342)

Compared with the actual social situation in Hebei Province, it is found that from the implementation of the tax reform policy in July 2016 to 2020, the cumulative reduction of groundwater overexploitation in Hebei Province is 4.35 billion m3, the ecological water consumption has increased significantly, and the downward trend of groundwater has been effectively alleviated. Promoting the water resources tax reform policy can effectively promote the regional water-saving effect, and the water ecosystem restoration continues. In addition, the levy of water resources tax led to increased production costs. The major high-water consumption enterprises continue to introduce advanced water-saving equipment and technology, and strengthen water-saving management. The industrial water repetition rate in Hebei Province reached 90.8 % in 2020. This also proves that the simulation results of the implementation of the water resources tax reform policy are consistent with reality. (Manuscript: L433-442)

As the standard of water resources tax levied on residents' water use in Hebei Province is ‘Tax and fee for translation’, the residents of water resources tax will not have a direct impact. However, from the actual situation in China, improving water resources tax collection standards is inevitable. Therefore, the analysis of this part can be used for reference to re-formulate the standards of residential water resources tax in the future. (Manuscript: L488-492)

Compared with the actual economic situation of Hebei Province, it is found that since the water resources tax was collected, the investment volume, import and export volume, residents and enterprise income of Hebei Province have increased year by year. Still the growth rate decreased compared to before the tax reform. Because in the real social development, many factors affect the investment volume, import and export volume, residents, and enterprise income. Therefore, to a certain extent, it can offset the negative impact of the collection of water resources tax on economic growth and import and export volume, which is also the reason why the level of social and economic development can gradually return to a steady state under the impact of water resources tax. Compared with the water resources fee, levying the water resources tax increases Hebei provincial government revenue. By 2019, the province's cumulative income from water resources tax was 6.941 billion yuan, with an average monthly income of 165 million yuan, which was more than twice the average monthly income from water resources fees in 2015. This also shows that collecting water resources tax can effectively increase government revenue. Still, its tax revenue is not transferred to residents and enterprises, which also affects the economic and social development of Hebei Province to a certain extent. This also proves that the simulation results of the implementation of the water resources tax reform policy are consistent with reality. To make the level of social and economic development return to a steady state as soon as possible, it is necessary to earmark the tax collected and transfer the corresponding payment to residents and enterprises. (Manuscript: L523-539)

8. The discussion section lacks any kind of comparison with other studies (domestic and foreign) and fails to summarize and contextualize the findings.

Response: According to the suggestion of this reviewer, in the discussion section, we summarized the research background and results accordingly (L541-551 of the manuscript). In addition, we also add a comparison of other studies with this paper. Some kinds of literature are added to support the results of this study. (L552-628 of the manuscript).

The contradiction between the shortage of water resources and the increasing demand for water resources has become one of the fundamental reasons that hinder the sustainable development of the social economy (Kong et al., 2021). Optimizing water resource allocation through economic leverage and promoting regional water conservation has gradually become an effective water resources management method (Wang et al., 2010). The successful experience of 10 pilot provinces of water resources tax in China also proves that the water resources tax reform policy helps promote the rational utilization and systematic management of water resources in China (Ouyang et al., 2022). Through the quantitative simulation analysis of the water resources tax reform policy, it can also be found that the collection of water resources tax has an obvious incentive effect on the water-saving behavior of social water users. It also significantly impacts the comprehensive development of residents, enterprises, and the social economy. (Manuscript: L541-551)

In addition, we also modify the grammatical structure of the whole paper in order to improve the quality of the paper.

Special thanks for your valuable comments!

Reviewer #2:

1. The author needs to emphasize the innovative value of this paper.

Response: Thanks for the reviewer's comments. For an academic paper, its innovation value is the most important part. Therefore, we summarize the innovative value of this paper and describe it in the paper. (L25-27, L121-125 of the manuscript).

The research results of this paper reveal the internal logic of the comprehensive impact of water resources tax on the economy and society and provide an important basis for the national promotion of tax reform policy. (Manuscript: L25-27)

The marginal contributions of this paper are as follows: (1) The internal logical structure of the impact of water resources tax reform policy on economic growth is elaborated, and further analyzed the transmission path of the effects of water resources tax on the economy on this basis. (2) A water resource-embedded DSGE model is constructed to analyze the policy effects of water resources tax reform dynamically. (Manuscript: L121-125)

2. The Introduction part should start from the phenomena and problems in practice and lead to the research problem.

Response: According to the suggestion of this reviewer, we further clarify in Section 2 what the ongoing reforms are and summarize the problems in practice. Thus leads to the research question of this paper. Critical revisions are shown below (see L127-146 of the manuscript for details).

In China, water resources are owned by the state. According to the Water Law, units and individuals who directly draw water resources from rivers, lakes, or underground shall apply to the water administrative department or the river basin management agency for a water drawing license and pay water resources fees. The water resources tax reform policy is to change the water resources fee to water resources tax, in the form of a tax, to practice the system of paid use of water resources (Zhang, 2019; Berbel et al., 2019). The promulgation of the Resource Tax Law of the People’s Republic of China also directly shows that the water resources tax is a kind of resource tax (Xiong et al., 2019). Therefore, the analysis of the effect of water resources tax reform policy conforms to the analysis paradigm of fiscal policy. In this paper, the policy effect of water resources tax refers to the economic impact of water resources tax and the combined effect of the change in the water consumption behavior of water users. Therefore, the analysis of the behavior of micro-actors directly affected by the water resources tax policy implementation can help to understand the transmission mechanism of water resources tax. It can help to understand why each actor chooses to conserve water resources and reduce unreasonable water demand through water resources tax. Moreover, it can help to reflect the way of achieving the structural change of water consumption and the restructuring in water abstraction by the differential tax rate. The analysis of the effect of the water resources tax policy can explain how the water resources tax can alleviate the contradiction between the supply and demand of water resources, change the water supply structure and promote industrial upgrading, as well as infer and argue its impact on economic development. This part analyzes the impact mechanism of water resources tax on residents' life, enterprise production, and social development and draws the impact mechanism diagram (Figure 1-3). (Manuscript: L127-146)

3. The literature review should reflect the value of this research

Response: According to the comments of the reviewer, we revised the literature review and emphasized the research value of this paper. Critical revisions are shown below (see L75-125 of the manuscript for details).

Reviewing the existing literature, most of the current studies on the effect of water resources tax policy are short-term or static, and lack of studies on the overall and lasting effects, such as using the CGE model to study the optimal tax rate on water resources (Tian et al., 2021), which has not yet formed the internal logical structure of the impact of water resources tax policies on economic growth. On the other hand, although the theoretical basis of the DSGE model can solve the above problems, there is a lack of a DSGE model on comprehensive water resources, and the available reference comes from the research results of the DSGE model on carbon taxation. Based on this, this paper constructs a DSGE model incorporating water resources based on the characteristics and economic value of water resources. Taking the first pilot tax reform in Hebei Province as an example to simulate the long-term dynamic response mechanism of water resources tax, to explore how the behavior of micro-entities makes decisions and to evaluate the effect of water resources tax on social water conservation and water resources protection from the perspective of long-term development. The marginal contributions of this paper are as follows: (1) The internal logical structure of the impact of water resources tax reform policy on economic growth is elaborated, and further analyzed the transmission path of the effects of water resources tax on the economy on this basis. (2) A water resource-embedded DSGE model is constructed to analyze the policy effects of water resources tax reform dynamically. (Manuscript: L110-125)

4. The innovation of this paper and the contribution made by previous studies have not been clearly expressed.

Response: According to the comments of the reviewer, we summarize the innovative value of this paper and describe it in the paper. (L25-27, L121-125 of the manuscript).

The research results of this paper reveal the internal logic of the comprehensive impact of water resources tax on the economy and society and provide an important basis for the national promotion of tax reform policy. (Manuscript: L25-27)

The marginal contributions of this paper are as follows: (1) The internal logical structure of the impact of water resources tax reform policy on economic growth is elaborated, and further analyzed the transmission path of the effects of water resources tax on the economy on this basis. (2) A water resource-embedded DSGE model is constructed to analyze the policy effects of water resources tax reform dynamically. (Manuscript: L121-125)

5. This article has obtained some interesting findings through the models, but these findings need to be further verified from theory or actual conditions. Also, further highlight the contribution of this article.

Response: The simulation results should be compared and analyzed with the actual situation further to illustrate the practical significance of the simulation results. Therefore, we have added the corresponding content in Section 4 (L433-442, L488-492, L523-539 of the manuscript).

Compared with the actual social situation in Hebei Province, it is found that from the implementation of the tax reform policy in July 2016 to 2020, the cumulative reduction of groundwater overexploitation in Hebei Province is 4.35 billion m3, the ecological water consumption has increased significantly, and the downward trend of groundwater has been effectively alleviated. Promoting the water resources tax reform policy can effectively promote the regional water-saving effect, and the water ecosystem restoration continues. In addition, the levy of water resources tax led to increased production costs. The major high-water consumption enterprises continue to introduce advanced water-saving equipment and technology, and strengthen water-saving management. The industrial water repetition rate in Hebei Province reached 90.8 % in 2020. This also proves that the simulation results of the implementation of the water resources tax reform policy are consistent with reality. (Manuscript: L433-442)

As the standard of water resources tax levied on residents' water use in Hebei Province is ‘Tax and fee for translation’, the residents of water resources tax will not have a direct impact. However, from the actual situation in China, improving water resources tax collection standards is inevitable. Therefore, the analysis of this part can be used for reference to re-formulate the standards of residential water resources tax in the future. (Manuscript: L488-492)

Compared with the actual economic situation of Hebei Province, it is found that since the water resources tax was collected, the investment volume, import and export volume, residents and enterprise income of Hebei Province have increased year by year. Still the growth rate decreased compared to before the tax reform. Because in the real social development, many factors affect the investment volume, import and export volume, residents, and enterprise income. Therefore, to a certain extent, it can offset the negative impact of the collection of water resources tax on economic growth and import and export volume, which is also the reason why the level of social and economic development can gradually return to a steady state under the impact of water resources tax. Compared with the water resources fee, levying the water resources tax increases Hebei provincial government revenue. By 2019, the province's cumulative income from water resources tax was 6.941 billion yuan, with an average monthly income of 165 million yuan, which was more than twice the average monthly income from water resources fees in 2015. This also shows that collecting water resources tax can effectively increase government revenue. Still, its tax revenue is not transferred to residents and enterprises, which also affects the economic and social development of Hebei Province to a certain extent. This also proves that the simulation results of the implementation of the water resources tax reform policy are consistent with reality. To make the level of social and economic development return to a steady state as soon as possible, it is necessary to earmark the tax collected and transfer the corresponding payment to residents and enterprises. (Manuscript: L523-539)

6. Compared with the available literature, what are the theoretical contributions and application values of this study? It is suggested to enhance the corresponding discussions in the conclusion part.

Response: According to the reviewer's comments, we summarize this paper's theoretical contribution in the literature review section and strengthen the discussion of the contribution of this paper in the conclusion section. (L110-125, L630-647 of the manuscript).

Reviewing the existing literature, most of the current studies on the effect of water resources tax policy are short-term or static, and lack of studies on the overall and lasting effects, such as using the CGE model to study the optimal tax rate on water resources (Tian et al., 2021), which has not yet formed the internal logical structure of the impact of water resources tax policies on economic growth. On the other hand, although the theoretical basis of the DSGE model can solve the above problems, there is a lack of a DSGE model on comprehensive water resources, and the available reference comes from the research results of the DSGE model on carbon taxation. Based on this, this paper constructs a DSGE model incorporating water resources based on the characteristics and economic value of water resources. Taking the first pilot tax reform in Hebei Province as an example to simulate the long-term dynamic response mechanism of water resources tax, to explore how the behavior of micro-entities makes decisions and to evaluate the effect of water resources tax on social water conservation and water resources protection from the perspective of long-term development. The marginal contributions of this paper are as follows: (1) The internal logical structure of the impact of water resources tax reform policy on economic growth is elaborated, and further analyzed the transmission path of the effects of water resources tax on the economy on this basis. (2) A water resource-embedded DSGE model is constructed to analyze the policy effects of water resources tax reform dynamically. (Manuscript: L110-125)

As a price lever, water resources tax is an effective measure to implement the idea of ‘water saving priority’. On the one hand, the levy of a water resources tax can change the unreasonable behavior of water use and force enterprises to improve production methods to improve water use efficiency. On the other hand, it can also reduce the dual pressure of the economy and environment so that consumers can establish a correct view of resources and protect water resources spontaneously and consciously. By deconstructing the internal logic of the impact of the water resources tax reform policy on economic growth, this paper puts forward the transmission path of the effects of water resources tax on finances. On this basis, the water resources elements are included in the DSGE model for the first time. The impact of water resources tax on micro-sector and socially sustainable development is simulated through impulse response analysis. It was found that, on the one hand, the water resources tax can effectively achieve the water conservation goal and help promote water-saving production, change the industrial water use structure, and improve water use efficiency. On the other hand, the water resources tax can help raise awareness of water conservation among enterprises and residents. The adjustment of the price mechanism is helpful to encourage enterprises to change the way of water intake and optimize the production structure. In addition, the effective implementation of the water resources tax is guaranteed by the reasonable and efficient use of special water resources protection funds. In this way, the water ecological environment can be effectively protected, and the recycling capacity of water resources and water supply capacity can be improved to meet the water demand. (Manuscript: L630-647)

7. The following literature should be helpful for your research：（1）Decoupling economic growth from water consumption in the Yangtze River Economic Belt, China.（2）Coordination of the Industrial-Ecological Economy in the Yangtze River Economic Belt, China.

Response: Thank you for the referees to provide references. These kinds of literature are beneficial for us to improve our research. We also cite these references. (L38-41, L541-543 of the manuscript).

In addition, we also modify the grammatical structure of the whole paper in order to improve the quality of the paper.

Special thanks for your good comments.

Reviewer #3:

1. The paper is an interesting study and a timely research work, presenting analysis of the behavior of micro-actors directly affected by the implementation of water resources tax policy that can help to understand and reduce unreasonable water demand through water resources tax. It is a complete work; well written and structured with extensive literature review and comprehensive analyses. The authors should however proofread their paper once more as some parts are still not very clear or need further improvements. Otherwise, excellent contribution to the body of knowledge in the respective field.

Response: Thanks for your affirmation of our study. We are also aware of some problems that remain in the article. Therefore, we revised the paper from the perspective of the proposal of the research question, the innovative value of the research, and the relevant discussion of the research conclusion. In addition, we also modify the grammatical structure of the whole paper in order to improve the quality of the paper.

Special thanks for your good comments.

Reviewer #4:

1. I found very poor English and sentence structuring throughout the paper. Hence extensive English editing may be required. (1) For example, use of “and” in the first line of abstract more than thrice. (2) Extraordinary long sentences misleading the true meaning of what the authors are trying to explain. (3) Study aim / need of study should be clear in the abstract. (4) The objectives of the study are not narrated in proper English. (5) It is good if the abstract become a bit short rather such long as the reader may get bored.

Response: Thank you for your comments. We are also aware that the English writing of the paper is inferior. Therefore, we focus on revising the grammar and structure of the form when revising the essay (full text of the manuscript). Reduce the use of long and complex sentences, and try to make the paper concise and clear. We have also changed the abstract to make it short and concise. At the same time, the purpose and significance of the research are also clarified in the abstract (L12-27 of the manuscript).

Abstract: Whether the implementation of the water resources tax policy can stimulate the water-saving behavior of social water users is one of the important criteria for evaluating the implementation effect of the tax reform policy. Taking Hebei Province, the first tax reform pilot in China, as an example. A dynamic stochastic general equilibrium model (DSGE) with embedded water resources tax is constructed to simulate the persistent impact of water resources tax on water-saving objectives. The research shows that: (1) Water resources tax can effectively achieve the goal of water-saving and improve the utilization efficiency of water resources. (2) Levying water resources tax helps to improve the water-saving awareness of enterprises and residents. It can also encourage enterprises to optimize production structures. (3) Rational and efficient use of special water resources protection funds is the basis for ensuring the effective implementation of water resources tax. It can also improve the recycling capacity of water resources. The results show that the government should speed up formulating a reasonable water resources tax rate and accelerate the construction of water resources tax protection measures. To ensure the relatively steady state of water resources utilization and protection, and achieve the dual goals of sustainable economic development and sustainable use of water resources. The research results of this paper reveal the internal logic of the comprehensive impact of water resources tax on the economy and society and provide an important basis for the national promotion of tax reform policy. (Manuscript: L12-27)

2. Keywords may be changed

Response: Thank you for your comments. We modify the keywords based on combing the full text (L30 of the manuscript).

Key words: Water resources tax, Water saving behavior, DSGE model, Policy effect, Hebei Province (Manuscript: L30)

3. The theoretical framework and the methodology are the most important and well elaborative part of the study and I want to appreciate the author for such deep insight

Response: Thank you for your recognition of our research. The analysis of the theoretical framework is a contribution of this paper. There are no references in the initial manuscript to support our argument. Therefore, we modified this part of the content when we revise the paper (L126-227 of the manuscript).

4. I think there is a lack of references in the discussion section.

Response: Thank you for your comments. We also note the lack of references in the discussion section. Therefore, we added 12 references to the discussion section when revising the paper to improve the deficiency of the discussion section (L540-628 of the manuscript).

5. All the results should be balanced with some references with prior studies and may be discussed in the discussion section.

Response: According to the suggestion of this reviewer, in the discussion section, we summarized the research background and results accordingly (L541-551 of the manuscript). In addition, we also add a comparison of other studies with this paper. Some kinds of literature are added to support the results of this study. (L552-628 of the manuscript).

The contradiction between the shortage of water resources and the increasing demand for water resources has become one of the fundamental reasons that hinder the sustainable development of the social economy (Kong et al., 2021). Optimizing water resource allocation through economic leverage and promoting regional water conservation has gradually become an effective water resources management method (Wang et al., 2010). The successful experience of 10 pilot provinces of water resources tax in China also proves that the water resources tax reform policy helps promote the rational utilization and systematic management of water resources in China (Ouyang et al., 2022). Through the quantitative simulation analysis of the water resources tax reform policy, it can also be found that the collection of water resources tax has an obvious incentive effect on the water-saving behavior of social water users. It also significantly impacts the comprehensive development of residents, enterprises, and the social economy. (Manuscript: L541-551)

Special thanks for your good comments.

Thank you and best regards.

Yours sincerely,

Guiliang Tian

tianguiliang@hhu.edu.cn

---

## [Decision Letter · Decision Letter 1]

5 Feb 2023

Can the water resources tax policy effectively stimulate the water saving behavior of social water users? A DSGE model embedded in water resources tax

PONE-D-22-20549R1

Dear Dr. Tian,

We’re pleased to inform you that your manuscript has been judged scientifically suitable for publication and will be formally accepted for publication once it meets all outstanding technical requirements.

Kind regards,

Abdul Majeed

Academic Editor

PLOS ONE

Additional Editor Comments (optional):

Reviewers' comments:

Reviewer's Responses to Questions

**Comments to the Author**

1. If the authors have adequately addressed your comments raised in a previous round of review and you feel that this manuscript is now acceptable for publication, you may indicate that here to bypass the “Comments to the Author” section, enter your conflict of interest statement in the “Confidential to Editor” section, and submit your "Accept" recommendation.

Reviewer #1: All comments have been addressed

Reviewer #3: All comments have been addressed

2. Is the manuscript technically sound, and do the data support the conclusions?

Reviewer #1: Yes

Reviewer #3: Yes

3. Has the statistical analysis been performed appropriately and rigorously? 

Reviewer #1: Yes

Reviewer #3: N/A

4. Have the authors made all data underlying the findings in their manuscript fully available?

Reviewer #1: Yes

Reviewer #3: Yes

5. Is the manuscript presented in an intelligible fashion and written in standard English?

Reviewer #1: Yes

Reviewer #3: Yes

6. Review Comments to the Author

Reviewer #1: Authors have addressed all comments, and the quality of manuscript has been improved. I am pleased to recommend the publication of the article.

Reviewer #3: Both the responses and revisions to the feedback by the reviewers seems to be sufficiently addressed, and therefore I would like to thank the authors for their contribution.

7. PLOS authors have the option to publish the peer review history of their article (what does this mean?). If published, this will include your full peer review and any attached files.

Reviewer #1: No

Reviewer #3: **Yes: **Hasim Altan
